# Ultrastructural Pathology of Atherosclerosis, Calcific Aortic Valve Disease, and Bioprosthetic Heart Valve Degeneration: Commonalities and Differences

**DOI:** 10.3390/ijms21207434

**Published:** 2020-10-09

**Authors:** Alexander Kostyunin, Rinat Mukhamadiyarov, Tatiana Glushkova, Leo Bogdanov, Daria Shishkova, Nikolay Osyaev, Evgeniy Ovcharenko, Anton Kutikhin

**Affiliations:** Department of Experimental Medicine, Research Institute for Complex Issues of Cardiovascular Diseases, 6 Sosnovy Boulevard, 650002 Kemerovo, Russia; kostae@kemcardio.ru (A.K.); muhara@kemcardio.ru (R.M.); glushtv@kemcardio.ru (T.G.); bogdla@kemcardio.ru (L.B.); shidk@kemcardio.ru (D.S.); osyaev@kemcardio.ru (N.O.); ovchea@kemcardio.ru (E.O.)

**Keywords:** atherosclerosis, calcific aortic valve disease, calcific aortic stenosis, bioprosthetic heart valves, structural valve deterioration, structural valve degeneration, calcification, inflammation, neovascularisation, electron microscopy

## Abstract

Atherosclerosis, calcific aortic valve disease (CAVD), and bioprosthetic heart valve degeneration (alternatively termed structural valve deterioration, SVD) represent three diseases affecting distinct components of the circulatory system and their substitutes, yet sharing multiple risk factors and commonly leading to the extraskeletal calcification. Whereas the histopathology of the mentioned disorders is well-described, their ultrastructural pathology is largely obscure due to the lack of appropriate investigation techniques. Employing an original method for sample preparation and the electron microscopy visualisation of calcified cardiovascular tissues, here we revisited the ultrastructural features of lipid retention, macrophage infiltration, intraplaque/intraleaflet haemorrhage, and calcification which are common or unique for the indicated types of cardiovascular disease. Atherosclerotic plaques were notable for the massive accumulation of lipids in the extracellular matrix (ECM), abundant macrophage content, and pronounced neovascularisation associated with blood leakage and calcium deposition. In contrast, CAVD and SVD generally did not require vasculo- or angiogenesis to occur, instead relying on fatigue-induced ECM degradation and the concurrent migration of immune cells. Unlike native tissues, bioprosthetic heart valves contained numerous specialised macrophages and were not capable of the regeneration that underscores ECM integrity as a pivotal factor for SVD prevention. While atherosclerosis, CAVD, and SVD show similar pathogenesis patterns, these disorders demonstrate considerable ultrastructural differences.

## 1. Introduction

Atherosclerosis and calcific aortic valve disease (CAVD) represent chronic inflammatory disorders which are characterised by endothelial dysfunction, lipid deposition, macrophage infiltration, and the maladaptive remodelling of the extracellular matrix (ECM) in the arterial wall and valve leaflets [1,2,3,4,5]. The altered paracrine signalling of the dysfunctional endothelium and blood-derived immune cells results in the excessive release of extracellular proteases, as well as pro-inflammatory and pro-calcific cytokines and growth factors [1,2,3,4,5]. The pathological microenvironment also induces the proliferation, phenotypic switch, and osteochondrogenic differentiation of vascular smooth muscle cells (VSMCs) and valvular interstitial cells (VICs), which additionally disturbs the balance between ECM deposition and degradation, concurrently promoting the calcification of the degraded ECM [1,2,3,4,5]. In arteries, established atherosclerotic lesions (i.e., plaques) disrupt the blood flow and may occlude the blood vessel after their erosion, rupture, or vasospasm, while the progressing degeneration and mineralisation of valve leaflets results in their mechanical incompetence, which leads to valve stenosis or insufficiency, ventricular hypertrophy, and heart failure [1,2,3,4,5].

Bioprosthetic heart valves (BHVs) are fabricated from bovine/porcine pericardium or porcine heart valves and undergo structural deterioration (SVD) over time due to the lack of a regenerative capability upon chemical fixation [6,7,8]. In most of these cases, SVD is a consequence of ECM disintegration, fibrosis, and calcification, all provoked by haemodynamic stress and the infiltration of the immunogenic prosthetic tissue by the macrophages of the recipient. If uncontrolled, SVD critically impairs the functioning of the implanted valve, causing recurrent stenosis or insufficiency [6,7,8]. In spite of the current advances in the design and manufacturing of BHVs, SVD remains a common long-term complication [9,10,11] demanding a repeated valve replacement in half of the patients 15 years post-implantation [12].

In agreement with the common mechanisms of development, atherosclerosis, CAVD, and SVD share a number of risk factors, including smoking, arterial hypertension, dyslipidaemia, diabetes mellitus, and chronic kidney disease [13,14,15,16]. These triggers, in particular when combined in comorbid conditions which are frequent in elderly, disrupt vascular and valvular homeostasis through multiple cellular and molecular pathways, which results in numerous histopathological patterns. Neovascularisation and calcification are among their most prominent features; yet, existing histological techniques include sectioning, which causes irreversible damage to the fresh, paraffin-embedded, or snap-frozen samples with ectopic mineral deposits, and often fail to provide high-quality images for the proper analysis of microcirculation. Therefore, the ultrastructural comparison of calcified vascular and valvular tissues remains an unmet pathophysiological need to better understand the pathological scenarios underlying atherosclerosis, CAVD, and SVD. If accomplished, this might improve the prevention and treatment approaches which currently target these disorders separately and do not aim for a combined reduction in vascular and valvular risk.

With the aim to solve this issue, we have previously developed an original protocol for the staining, embedding, and backscattered scanning electron microscopy visualisation of the mineralised cardiovascular specimens [17,18,19] which entirely retains the integrity of the calcified tissues and combines the advantages of light microscopy and transmission electron microscopy, providing high-resolution images and ultrastructural details of all the vascular and valvular tissue structures and cell types. The sample preparation is straightforward but requires a machine for the grinding and polishing (e.g., TegraPol by Struers). The consecutive grinding of the sample permits layer-by-layer scanning that is principally similar to the sectioning but does not cause any damage to the mineralised tissues. Although backscattered scanning electron microscopy was earlier utilised for the investigation of calcium phosphate transformation in human samples [20,21,22,23], it has never been applied in cardiovascular pathology.

Here, we employed the above-mentioned approach for performing an ultrastructural investigation of atherosclerotic plaques, calcified native aortic valves (AVs), and degenerated BHVs to compare the pathological phenomena characteristic of the inflamed cardiovascular tissues. Among them were lipid retention, macrophage infiltration and specialisation, neovascularisation, intraplaque/intraleaflet haemorrhage, and mineralisation. Lipid retention, haemorrhages, and calcium deposits were frequently detected across plaques and affected valves. However, intraplaque haemorrhages were associated with leaky neovessels, whereas intraleaflet haemorrhages were provoked by the tissue delamination. Abundant neovascularisation was specific for plaques, while macrophage diversity was a feature of SVD. 

## 2. Results

We first performed a semi-quantitative analysis of all the samples (36 atherosclerotic plaques, 12 calcified native AVs, and 12 failed BHVs with SVD). As underlying health conditions could impact on the development of histopathological features, we have also analysed the prevalence of the patient comorbidities, but did not find any statistically significant differences between the groups (Table 1).

Each of the disorders has been characterised by a specific histopathological pattern (Table 2). Foam cells were detected in all plaques but were less frequent in diseased native AVs and BHVs. Canonical macrophages (i.e., mononuclear macrophages without any cytoplasmic inclusions) were encountered regardless of the tissue, whereas multinucleated giant cells were characteristic of failed BHVs. Neutrophil infiltration was often noted in plaques and BHVs, but not in the native AVs. Endothelialisation or pseudoendothelialisation (a layer of endothelial-like cells at the surface) were common for all the studied specimens. Neovascularisation was particularly notable in plaques. While haemorrhages were observed across all pathologies, they were caused by the leakage of neovessels in plaques and by red blood cell (RBC) penetration or delamination in native AVs and BHVs. Mineral deposits were detected in almost all the examined samples.

We further carried out a detailed analysis of the indicated ultrastructural features. As atherosclerosis, CAVD, and SVD all involve macrophage infiltration and are largely driven by lipid accumulation, we primarily focused on the formation of foam cells. Depending on the dehydration efficiency, they appeared as large round, oval, or irregularly shaped cells full of black or white globules (Figure 1A). Among all the examined cardiovascular tissues, foam cells were the most abundant in the atherosclerotic plaques, where they generally located as multiple aggregations randomly distributed across the neointima (Figure 1A). Native AVs and BHVs contained a few lipid-laden cells (Figure 1A), although fatty streaks were also prominent at the inflow and outflow surfaces (Figure 1A,B).

We next investigated the histopathological diversity of the macrophages in the affected vascular and valvular tissues. The majority of the macrophages had elliptic or round large nuclei and did not have any specific inclusions in the cytosol, suggesting their moderate activity in the ECM remodelling (Figure 2A). However, some of the valvular macrophages, typically located within the degraded ECM, contained numerous electron-dense granules and had an intermediate phenotype between canonical macrophages and multinucleated giant cells (Figure 2B). This macrophage subtype was especially prominent in BHVs (Figure 2B). Multinucleated giant cells with multiple cytoplasmic inclusions were detectable in the atherosclerotic plaques and BHVs (Figure 2C). Besides the macrophages, the plaques and BHVs were infiltrated by neutrophils, suggestive of active inflammation (Figure 2D).

Monocyte infiltration followed by a macrophage-driven ECM disintegration represents a multi-step process. Electron microscopy snapshots enabled to illustrate its events, including rolling (Figure 3A), adhesion (Figure 3B), the invasion of the cells into the ECM (Figure 3C,D), the degradation of the surrounding ECM and engulfment of its components (Figure 3E), and ECM delamination (Figure 3F).

Leukocyte rolling and attachment are induced by endothelial activation and the loss of endothelial integrity, although endothelial dysfunction does not necessarily mean endothelial injury, and re-endothelialisation is a common regenerative phenomenon. Atherosclerotic plaques and native AVs were covered by a layer of endothelial cells elongated along the direction of flow, while BHVs had single endothelial-like cells attached to the prosthetic surface (Figure 4A). Mesenchymal cells in the examined cardiovascular tissues mostly included vascular/valvular SMCs or VICs (Figure 4B). Fibroblasts were rarely encountered (Figure 4C).

We found intraplaque haemorrhage as a frequent event in the atherosclerotic plaques because of leaky neovessels (Figure 4D). In contrast to the vasa plaquorum, microvessels within the native AVs and capillary-like tubes in the BHVs were impermeable for red blood cells (RBCs) (Figure 4D). Yet, both types of valve were also penetrated by RBCs, which was accompanied by impaired ECM integrity (Figure 4E) as well as degradation (Figure 5A) and the disorientation of the collagen fibers (Figure 5B). The disintegration of the ECM was associated with haemorrhagic infiltration (Figure 5C), the delamination of the tissue (Figure 5D), and the migration of RBCs into the hollows (Figure 5E).

Mineral deposits were characterised by a remarkable heterogeneity. Both plaques and native/bioprosthetic valves contained macrocalcifications between the inflow and outflow surfaces (Figure 6A), which could be surrounded (plaques and native AVs) or covered (BHVs) by a connective tissue (Figure 6B). Some of the calcifications had uneven or sharp edges, suggestive of its invasive pattern (Figure 6C). Multiple microcalcifications of varying diameter were often detected in all the examined tissues (Figure 6D).

## 3. Discussion

The concept that the development of atherosclerosis and heart valve disease is interrelated was a matter of debate during the last decade [24,25,26,27]. The arguments include common genetic susceptibility loci [28,29,30] and a number of identical circulating biomarkers [31,32,33]; yet, the similarities and differences in the pathophysiology of atherosclerosis, CAVD, and structural degeneration of BHVs have not been systematically compared. In particular, there are a lack of ultrastructural data connecting the pathophysiology of these disorders with their clinical manifestations. The sample preparation of extraskeletal calcified tissues such as plaques or dysfunctional heart valves is complicated, since it is incompatible with histological sectioning. Having applied our original protocol for the staining, embedding, and visualisation of the mineralised cardiovascular specimens, here we carried out an electron microscopy investigation of the main histopathological features of atherosclerosis, CAVD, and SVD—i.e., macrophage phenotypic switch and infiltration, intraplaque/intraleaflet haemorrhage, ECM degradation, and calcification.

Despite lipid deposition being prominent regardless of the tissue type, it was most pronounced in atherosclerotic plaques where the foam cells formed multiple conglomerates in contrast to native and prosthetic valves. However, the macrophage diversity in plaques (and also in calcified native AVs) was limited, whereas BHVs contained a number of macrophage subtypes (foam cells, canonical macrophages, mononucleated macrophages with multiple cytoplasmic inclusions, and multinucleated giant cells). The blood vessels in plaques and native AVs had distinct structural properties. Plaque neovessels frequently leaked and were responsible for intraplaque haemorrhages. While native AVs also relied on the microvasculature to meet the nutritional needs, it generally did not leak and intraleaflet haemorrhages have been associated with stress-driven ECM degradation and delamination. Substantial delamination and ECM disintegration were characteristic of BHVs which were incapable of regeneration. The calcification patterns in the arteries and heart valves were similar and included both large mineral deposits and microcalcifications. The schematic representation of the study findings is depicted in Figure 7.

Macrophage infiltration is evident in both atherosclerotic and valvular disease, being responsible for the neointimal and dystrophic remodeling, respectively [34,35,36,37]. The phenotyping of macrophages is mostly based on immunodetection methods such as immunostaining or flow cytometry upon fluorescence- or magnetic-activated cell sorting. Briefly, all macrophages are differentiated into pro-inflammatory (M1) and anti-inflammatory (M2), although this classification is oversimplified (at least for in vivo scenarios), and macrophages demonstrate a remarkable plasticity depending on the microenvironmental cues [38,39,40,41]. Although being widely established, this approach should ideally be combined with the ultrastructural interrogation of macrophages which also permits to identify their location and co-localisation with other plaque or valvular compartments. The current macrophage hierarchy does not consider their ECM remodelling activity, which is of crucial importance in the development of atherosclerosis, CAVD, and SVD [1,2,3,5,42], while backscattered electron microscopy permits the high-quality visualisation of cellular content and allows us to evaluate the integrity of the surrounding ECM. 

Recent single-cell sequencing studies have indicated the leading role of non-lipid-laden macrophages in promoting the neointimal inflammation [43,44,45]. In our study, non-lipid-laden macrophages with numerous electron-dense granules and multinucleated giant cells were located in the areas with degraded ECM that might suggest their pro-inflammatory role. The appreciable diversity of ultrastructural macrophage phenotypes in BHVs in comparison with native cardiovascular tissues is probably explained by the amount of the heterologous biomaterial possessing residual immunogenicity, which is caused by galactose-α-1,3-galactose and N-glycolylneuraminic acid, essential components of bovine and porcine ECM that have been lost by humans during evolution [46]. These ultrastructural findings support the current vision on immune rejection as one the main factors contributing to the BHV failure [42]. In turn, the excess the lipid-laden macrophages in the plaques reflects the paramount role of lipid retention in atherosclerosis, whereas in dysfunctional heart valves it merely complements the mechanical stress and immune rejection mechanisms, though still being among the major contributors.

(Neo)vascularisation provides a route for the migration of macrophages and other immune cells to the degraded ECM and altered microenvironment, thereby promoting inflammation and calcification in plaques and AVs [47,48,49,50]. While neovessels were detected in both plaques and AVs, plaque vessels were immature and leaky and often resulted in intraplaque haemorrhage, while valvular neovessels retained the integrity of the endothelial barrier. Nevertheless, haemorrhages were also common for the native AVs and BHVs because of ECM disintegration and delamination upon the critical valve fatigue [51]. Our results contradict some previously published studies which reported that microvascular leakage is a frequent phenomenon in CAVD [52,53], and this discrepancy warrants further investigation. The disorganisation of the ECM meshwork was notable in diseased native AVs and especially in the BHVs. Combined with intraleaflet haemorrhage, this provoked tissue swelling and delamination/pseudoaneurysm, further disrupting the distribution of the mechanical load and impairing the biomechanical properties of the valve. In addition, local haemorrhages may enhance the development of the valvular and vascular inflammation through iron oxidation and the subsequent generation of reactive oxygen [54] and nitrogen [55] species.

The ultimate outcome of inflammation, mechanical stress, and intraplaque or intraleaflet haemorrhage is calcification [56,57,58], which has a variety of patterns considerably affecting the disease course [59,60,61,62,63]. The dystrophic calcification of degraded ECM components and biomineralisation mediated by the osteochondrogenic differentiation of mesenchymal cells are two different modalities of valve and plaque ossification [1,56,57,58]. Besides its trigger, extraskeletal calcification pattern is largely defined by the ambient conditions. Whether the mineral deposit is smooth or sharp, large or small, amorphous or crystalline strongly depends on microenvironmental pH as well as amount and proportions of available mineral ions [1]. Currently, microcalcifications < 5 µm diameter are not treated as hazardous [60], while calcifications between 5 and 100 µm diameter contribute to the plaque rupture because of uneven stress distribution across the delaminated tissue [61,62,63] and those > 100 µm diameter stabilise the plaque [61,63]. Despite the controversies on the role of different calcification modalities and patterns, it is clear that all of them are encountered in plaques, stenotic valves, and BHVs.

Differences in the prevalence and localisation of different ultrastructural features in distinct cardiovascular tissues may be linked to the site- or stage-specific alterations. Seemingly, the deposition of lipids in the ECM and their internalisation by macrophages play a key role in both the initiation and progression of atherosclerosis, as foam cells are abundant across the entire neointima. However, the localisation of lipid-laden macrophages in dysfunctional native or bioprosthetic valves is typically restricted to their inflow or outflow surfaces, suggesting that, in these settings, lipid retention and foam cells are responsible mainly for the initiation and do not have a pivotal significance further.

The main shortcoming of this study is that is has been limited to the observational approach and did not include any interventions. Nevertheless, here we focused on drawing parallels between the main inflammation-, remodeling-, and calcification-related disorders of the circulatory system components that did not require interventional study. To conclude, atherosclerosis, CAVD, and BHV failure share general pathogenetic mechanisms, while almost every of them has specific nuances and ultrastructural features which deserve further investigation.

## 4. Materials and Methods

Atherosclerotic plaques (*n* = 36) were excised during carotid endarterectomy conducted because of chronic brain ischemia. Calcified native AVs (*n* = 12) and BHVs (PeriCor or UniLine, NeoCor, Kemerovo, Russian Federation, *n* = 12) were obtained from the patients who underwent primary heart valve replacement because of CAVD or repeated heart valve replacement due to SVD, respectively. In-stent restenosis was the exclusion criterion for the patients with carotid atherosclerosis, whereas bicuspid AV, rheumatic heart disease, and infective endocarditis were exclusion criteria for those with CAVD. Median of BHV functioning was 5.75 (interquartile range 4.69–11.37) years. All surgical interventions were performed in the Research Institute for Complex Issues of Cardiovascular Diseases (Kemerovo, Russian Federation) in 2019–2020. The investigation was carried out in accordance with the Good Clinical Practice and the Declaration of Helsinki. The study protocol was approved by the Local Ethical Committee of Research Institute for Complex Issues of Cardiovascular Diseases (Protocol No. 20190606, date of approval 06 June 2019). All the patients provided a written informed consent after receiving a full explanation of the study.

Excised plaques as well as degenerated and calcified fragments of all available valve leaflets were fixed in two changes of 10% neutral phosphate buffered formalin (B06-003, BioVitrum, St. Petersburg, Russian Federation) for 24 h at 4 °C, postfixed in 1% osmium tetroxide (OsO_4_, 19110, Electron Microscopy Sciences, Hatfield, PA, USA) for 24 h, stained in 2% osmium tetroxide for 48 h, dehydrated in ascending ethanol series (50%, 60%, 70%, 80%, and 95%, 15 min per each), stained in 2% alcoholic uranyl acetate (22400-2, Electron Microscopy Sciences, Hatfield, PA, USA) for 5 h, dehydrated in isopropanol (06-002, BioVitrum, St. Petersburg, Russian Federation) for 5 h and acetone (1 h), impregnated with acetone: epoxy resin (Epon, 14120, Electron Microscopy Sciences, Hatfield, PA, USA) mixture (1:1) for 6 h and with epoxy resin for 24 h, and were finally embedded into the fresh epoxy resin at 60 °C. The samples were then grinded, polished (TegraPol-11, Struers, Copenhagen, Denmark), and counterstained with Reynolds’s lead citrate (17810, Electron Microscopy Sciences, Hatfield, PA, USA) for 7 min. After a brief washing in double distilled water, samples were sputter coated (10 nm thickness) with carbon (EM ACE200, Leica, Wetzlar, Germany) and visualised by means of backscattered scanning electron microscopy at 10 or 15 kV voltage (S-3400N, Hitachi, Tokyo, Japan).

For the lipid staining, the excised vessels and valves were snap-frozen in the optimal cutting temperature medium (Tissue-Tek, 4583, Sakura Finetek, Tokyo, Japan) and cut on a cryostat (Microm HM. Sections (5 μm thickness) were then fixed in 4% paraformaldehyde (P6148, Sigma-Aldrich, St. Louis, MO, USA) for 10 min, washed thrice in phosphate-buffered saline (P4417, Sigma-Aldrich, St. Louis, MO, USA), incubated in isopropanol for 5 min, stained with Oil Red O (ab150678, Abcam, Cambridge, UK) for 15 min, briefly washed in 60% isopropanol, counterstained with modified Mayer’s hematoxylin (ab150678, Abcam, Cambridge, UK), washed in tap water for 5 min and in double distilled water briefly, and finally mounted (Mowiol 4-88, 81381, Sigma-Aldrich, St. Louis, MO, USA). Visualisation was performed by light microscopy (AxioImager.A1, Carl Zeiss, Oberkochen, Germany).

Statistical analysis was performed using GraphPad Prism 7 (GraphPad Software, San Diego, CA, USA). Data were represented by the proportions or by the median and interquartile range. The proportions were compared by Pearson’s chi-squared test or Fisher’s exact test. *p* values ≤ 0.05 were regarded as statistically significant.

## Figures and Tables

**Figure 1 ijms-21-07434-f001:**
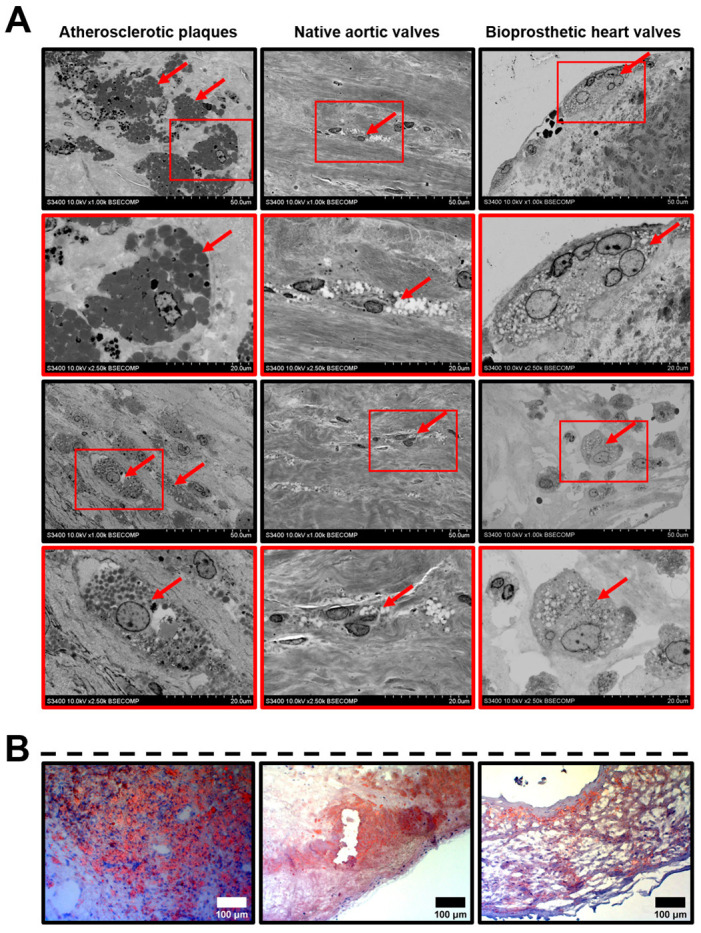
Lipid-laden macrophages (foam cells, indicated by red arrows) within the atherosclerotic plaques, calcified native AVs, and failed BHVs. (**A**) Different appearances of the lipid-laden macrophages depending on their location and phenotype, backscattered scanning electron microscopy, magnification 1000×–2500×. (**B**) Oil Red O staining of foam cells, magnification 200×. Note the multiple clusters of foam cells in the atherosclerotic plaques, as well as the fatty streaks beneath the surface of native AVs and BHVs, which, however, contained only single foam cells in the deep tissue layers.

**Figure 2 ijms-21-07434-f002:**
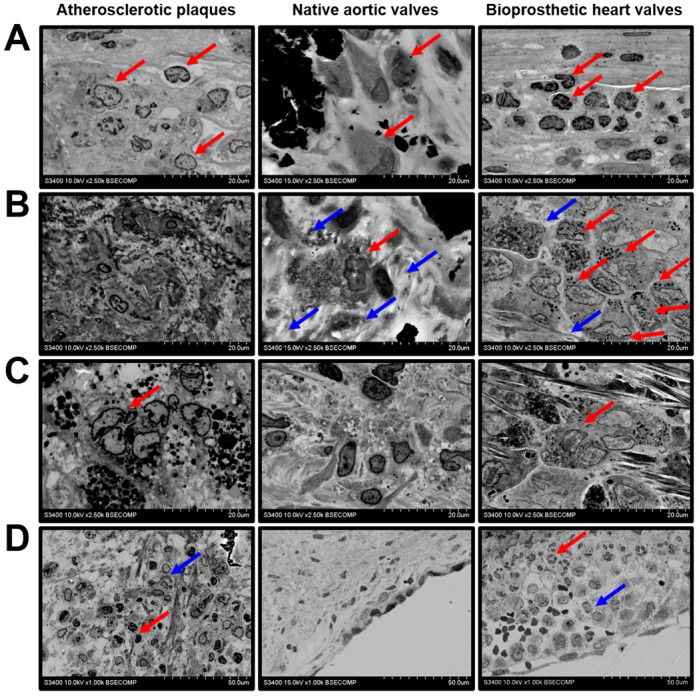
Macrophage diversity in the atherosclerotic plaques, calcified native AVs, and failed BHVs. (**A**) Canonical macrophages (indicated by red arrows) without cytoplasmic inclusions, magnification 2500×. (**B**) Macrophages with electron-dense cytoplasmic granules (indicated by red arrows), surrounded by the degraded ECM (indicated by blue arrows), magnification 2500×. Note the absence of this macrophage subtype in the atherosclerotic plaques. (**C**) Multinucleated giant cells with multiple electron-dense cytoplasmic inclusions (indicated by red arrows), magnification 2500×. Note the absence of this macrophage subtype in the native AVs. (**D**) Combined neutrophil (indicated by red arrows) and macrophage (indicated by blue arrows) infiltration, magnification 1000×. Note the absence of neutrophils in the native AVs.

**Figure 3 ijms-21-07434-f003:**
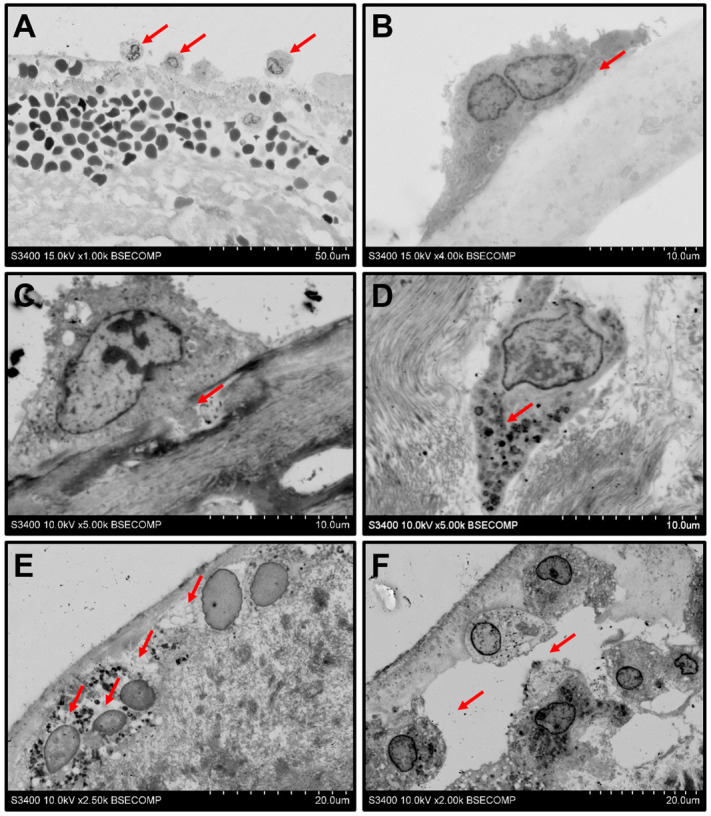
Events behind the monocyte infiltration of the BHVs. (**A**) Rolling of the monocytes (indicated by red arrows) along the BHV surface, magnification 1000×. (**B**) Adhesion to the surface (indicated by red arrows), magnification 4000×. (**C**) Cleavage of the ECM proteins and ECM degradation by invadopodia (indicated by red arrows), magnification 5000×. (**D**) Aggregation of the cytoplasmic granules at the leading edge of the macrophage (indicated by red arrows), magnification 5000×. (**E**) Degradation of the ECM around the macrophages (indicated by red arrows), magnification 2500×. (**F**) Delamination of the ECM upon its degradation (indicated by red arrows), magnification 2000×.

**Figure 4 ijms-21-07434-f004:**
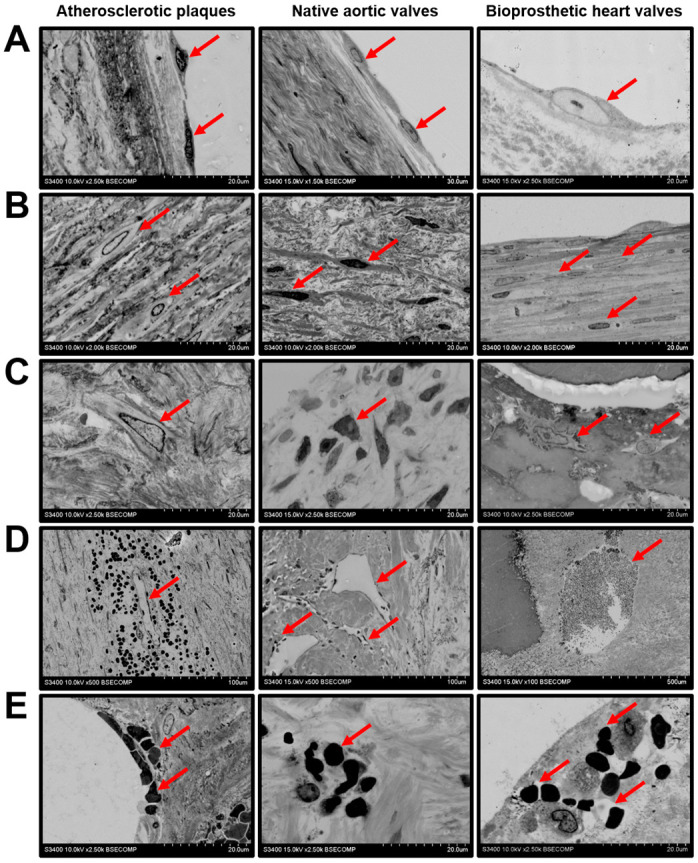
Cellular composition of the atherosclerotic plaques, calcified native AVs, and failed BHVs. (**A**) Endothelial and endothelial-like cells (indicated by red arrows), magnification 1500×–2500×. Note the endothelial monolayer in the atherosclerotic plaques and native AVs and single endothelial-like cells responsible for the pseudoendothelialisation of the BHVs. (**B**) Vascular/valvular SMCs and VICs (indicated by red arrows), magnification 2000×. Note the valvular SMCs within the BHV pannus. (**C**) Fibroblasts or fibroblast-like cells (indicated by red arrows), magnification 2500×. (**D**) Leaky neovessels within the atherosclerotic plaques, microvessels inside the native AVs, and capillary-like tubes in the BHVs (indicated by red arrows), magnification 100×–500×. (**E**) Intraplaque and intraleaflet microhaemorrhages (indicated by red arrows) occurring because of plaque neovessel leakage and the fatigue-induced degradation of the valvular ECM, respectively; magnification 2500×.

**Figure 5 ijms-21-07434-f005:**
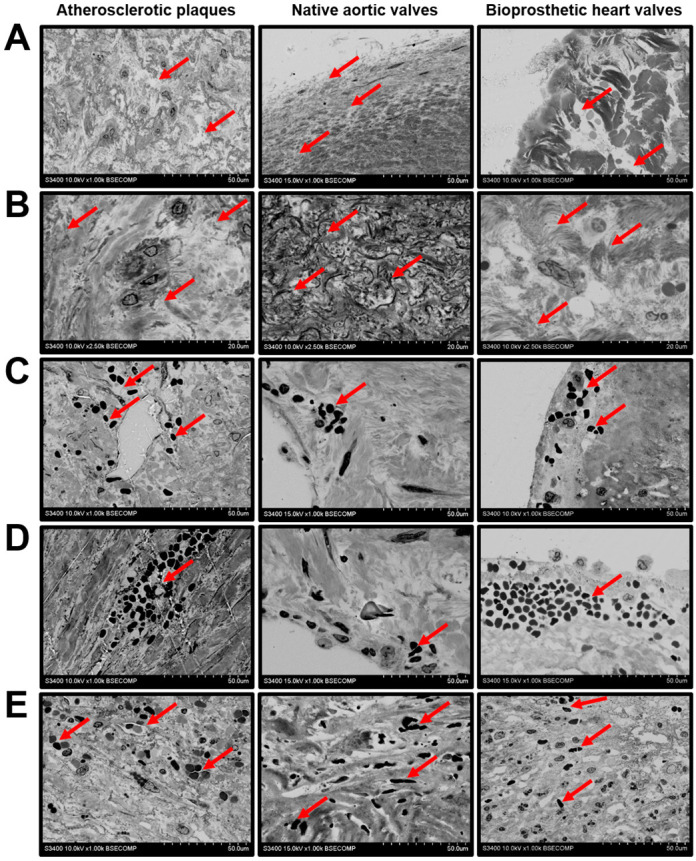
ECM degradation and haemorrhages in atherosclerotic plaques, calcified native AVs, and failed BHVs. (**A**) Loss of the ECM integrity upon the protease- and fatigue-mediated degradation of the ECM components (indicated by red arrows), magnification 1000×. (**B**) Loss of the ECM fiber orientation (indicated by red arrows), magnification 2500×. (**C**) Haemorrhagic infiltration (indicated by red arrows) because of leaky plaque neovessels or the stress-induced penetration of the inflow valve surface, magnification 1000×. (**D**) Delamination of the neointimal and valvular tissue (indicated by red arrows) upon the accumulation of blood in the cavities formed after the ECM degradation, magnification 1000×. (**E**) Haemorrhagic infiltration of deep ECM layers (indicated by red arrows) upon the massive intraplaque or intravalvular bleeding, magnification 1000×.

**Figure 6 ijms-21-07434-f006:**
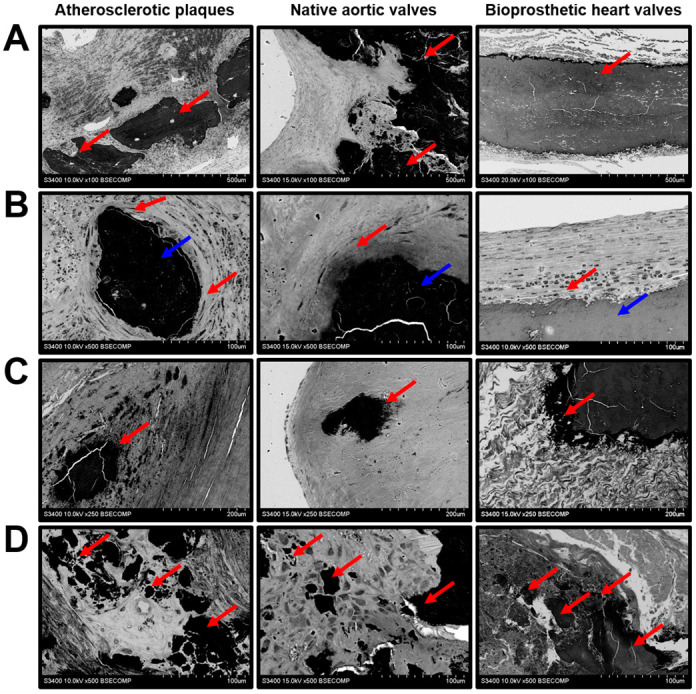
Mineralisation in the atherosclerotic plaques, calcified native AVs, and failed BHVs. (**A**) Macrocalcifications within the ECM (indicated by red arrows), magnification 100×. (**B**) Encapsulated macrocalcifications (capsule indicated by red arrows, calcifications indicated by blue arrows), magnification 500×. (**C**) Sharp-edged uneven calcifications (indicated by red arrows), magnification 250×. (**D**) The diversity of calcification patterns (indicated by red arrows), magnification 500×.

**Figure 7 ijms-21-07434-f007:**
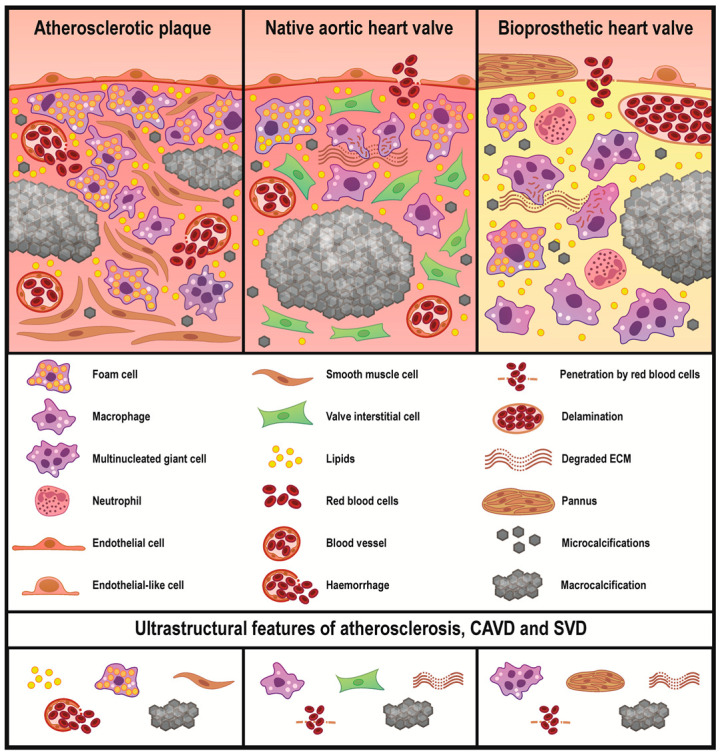
Common and unique ultrastructural features of atherosclerotic plaques, calcified native AVs, and failed BHVs. Atherosclerosis is mainly driven by a lipid retention accompanying by a foam cell formation. Subsequent migration and phenotypic switch of the vascular smooth muscle cells, active neovascularisation, and mineralisation further promote the development of atherosclerosis. Foam cells are also observed in the native AVs and BHVs, yet in considerably lower amounts, in particular when compared with non-lipid-laden macrophages. The highest macrophage diversity is observed in BHVs which additionally contain ECM-degrading/ECM-scavenging macrophages and multinucleated giant cells. In BHVs, mesenchymal cells form large outgrowths of a connective tissue which is termed pannus. Both heart valve types are characterised by protease- and fatigue-induced ECM degradation. Calcification and haemorrhages are common for all studied disorders, albeit intraplaque haemorrhages are caused by a neovessel leakage, while intravalvular bleedings are triggered by a loss of the ECM integrity.

**Table 1 ijms-21-07434-t001:** Clinicopathological features of the patients with atherosclerosis, CAVD, and BHV failure.

Feature/Sample	Patients with Atherosclerosis (*n* = 36)	Patient with CAVD (*n* = 12)	Patients with BHV Failure (SVD) (*n* = 12)	*p* Value
Male gender, *n* (%)	25 (69.4)	8 (66.7)	6 (50.0)	0.47
Age, years, Me (25th; 75th)	64 (58; 69)	63.5 (55; 66)	64.5 (61; 71.25)	0.92
Arterial hypertension, *n* (%)	34 (94.4)	10 (83.3)	10 (83.3)	0.37
Chronic heart failure, *n* (%)	34 (94.4)	12 (100.0)	12 (100.0)	0.50
COPD or asthma, *n* (%)	9 (25.0)	3 (25.0)	0 (0.0)	0.15
Chronic kidney disease, *n* (%)	8 (22.2)	4 (33.3)	7 (58.3)	0.07
Diabetes mellitus, *n* (%)	6 (16.7)	0 (0.0)	0 (0.0)	0.11
Overweight or obesity, *n* (%)	17 (47.2)	6 (50.0)	8 (66.7)	0.50

CAVD—calcific aortic valve disease; BHV—bioprosthetic heart valve; SVD—structural valve deterioration; Me—median, 25th; 75th— interquartile range; COPD—chronic obstructive pulmonary disease.

**Table 2 ijms-21-07434-t002:** Semi-quantitative analysis of the ultrastructural features in atherosclerotic plaques, calcified native AVs, and failed BHVs.

Feature/Sample	Atherosclerotic Plaques (*n* = 36)	Calcified Native AVs (*n* = 12)	Failed BHVs (*n* = 12)	*p* Value
Foam cells, *n* (%)	36 (100.0)	3 (25.0)	4 (33.3)	0.0001
Canonical macrophages, *n* (%)	36 (100.0)	12 (100.0)	12 (100.0)	0.99
Multinucleated giant cells, *n* (%)	8 (22.2)	0 (0.00)	9 (75.0)	0.0001
Neutrophil infiltration, *n* (%)	13 (36.1)	0 (0.00)	7 (58.3)	0.009
(Pseudo)endothelialisation, *n* (%)	26 (72.2)	12 (100.0)	6 (50.0)	0.02
Neovascularisation, *n* (%)	35 (97.2)	5 (41.7)	0 (0.0)	0.0001
Haemorrhages, *n* (%)	17 (47.2)	2 (16.7)	4 (33.3)	0.16
Leaky microvessels, *n* (%)	21 (58.3)	0 (0.00)	0 (0.0)	0.0001
Penetration by RBCs, *n* (%)	5 (13.9)	5 (41.7)	11 (91.7)	0.0001
Delamination, *n* (%)	4 (11.1)	2 (16.7)	9 (75.0)	0.0001
Microcalcification, *n* (%)	36 (100.0)	12 (100.0)	10 (83.3)	0.02
Macrocalcification, *n* (%)	36 (100.0)	12 (100.0)	10 (83.3)	0.02

AVs—aortic valves; BHVs—bioprosthetic heart valves; RBCs—red blood cells.

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
