# Peer review of "Ultrastructural Pathology of Atherosclerosis, Calcific Aortic Valve Disease, and Bioprosthetic Heart Valve Degeneration: Commonalities and Differences"

_ijms, 2020, doi:10.3390/ijms21207434_

Round 1
Reviewer 1 Report
The authors report a comparative observational study using a method to identify major histological features of disease including immune cells, cellular composition, extracellular matrix and ectopic mineralization at the ultrastructural level. This more detailed examination hints at differences in the disease progression at different sites. The study compared the ultrastructure of different types of cardiovascular disease and noted similarities and differences depending on the underlying etiology.
Some specific points:
Could an outline of patient characteristics (e.g. age, sex) for each group be included? This could be important as these factors are important risk factors in cardiovascular disease.
The abstract states “an original method for sample preparation and visualization” (lines 118-19). Can the authors address how this method is different, and why they believe this is an improvement? This would also allow the inclusion of some of the previous published ultrastructural studies that seem to be excluded.
Samples of atherosclerotic plaques, native and bioprosthetic valves were excised from multiple patients (lines 289-291). However, the data reported in the main body of the text is simply representative images of the different ultrastructural findings. Is there a way that the frequency of this findings can be quantified (for example observed in 12/12 valves or something similar)? This would provide the reader with some indication of the variability and strengthen the impact of the manuscript.
EM images are notoriously difficult to interpret for non-specialized readers, so is it possible to annotate the EM images throughout to highlight the features that the authors are describing?
In figure 1A, the second row of images shows higher magnification of the images in the 1st row, and same for rows 3 and 4. It might be beneficial to highlight that the ROI that is at higher magnification. Consistency throughout on the magnification of images (at least to enable comparison across the plaques, native and bioprosthetic valves) would be an improvement.
Figure 2, blank images are shown to represent that the features described are not present. Could these be replaced by images? This would allow readers that are not familiar to compare. Annotation (arrows) showing the features would allow the distinction between those features present / not.
Could the authors comment on the possibilities that the differences in ultrastructural findings at the different sites may be linked to either site specific or stage specific changes, in particular with regards to clarification of statement (line 89).
The summary diagram (Fig 7) is a brilliant schematic way to summarize the EM features described in the other figures and clearly show the differences in the cellular features. The inclusion of this brings the paper readability to a much wider audience.
Author Response
We sincerely thank the reviewer for the constructive criticism and valuable comments. We revised our manuscript according to the suggestions of the reviewer, which helped us to substantially improve the paper.
Please see the attachment.

Reviewer 2 Report
Using the original method based on backscattered scanning electron microscopy (BSEM) visualization, the authors have examined morphologically the most characteristic histopathologic features of atherosclerotic plaques, stenotic native aortic valves and bioprosthetic valves, which are lipid accumulation, macrophage infiltration, calcification and occurrence of haemorrhages. The involvement of inflammation, lipids and calcification in cardiovascular pathologies is well established. However, the lack of significant progress in prevention and therapy (in the case of aortic valve stenosis in particular) indicates that crucial mechanisms driving these socially important diseases are still not well understood. Hence, the efforts to clarify their nature and improve research methods “to look closer at the problem,” paraphrasing CM Otto (CM. Otto. Calcific Aortic Stenosis — Time to Look More Closely at the Valve. 2008).
Critical remarks
In general
As stated by the authors, the important limitation of this study is its strictly observational character (line 282). For this reviewer, more important is a descriptive form of the presentation of the results, not supported by more detailed and reliable, e.g. semiquantitative analysis.
Moreover, the authors find nothing so new and/or unexpected, which would have to justify such a brief analysis.
Lack of novelty is the second concern regarding this study. The authors conclusion - “To conclude, atherosclerosis, heart valve disease, and BHV failure share general pathogenetic mechanisms, while almost every of them has specific nuances and ultrastructural features which deserve further investigation.” (lines 285-287) – does not bring anything new to our knowledge about “the pathological scenarios underlying atherosclerosis, CAVD and SVD” (line 68).
What is new in this study (at least for this reviewer) is the interesting, original technique (BSEM) employed to examine the tissue material. However, as mentioned earlier, the use of this method does not imply here new findings not yet documented by other modalities (e.g. histochemistry, immunohistochemistry, SEM, TEM). Let me provide some, first in line, references to confirm my statement:
Bobryshev YV. Monocyte recruitment and foam cell formation in atherosclerosis. Micron. 2006;37(3):208-22.
Akahori H et al. Intraleaflet haemorrhage is associated with rapid progression of degenerative aortic valve stenosis. Eur Heart J. 2011 Apr;32(7):888-96.
Finn AV et al. Hemoglobin directs macrophage differentiation and prevents foam cell formation in human atherosclerotic plaques. J Am Coll Cardiol. 2012 Jan 10;59(2):166-77.
Ley K, Miller YI, Hedrick CC. Monocyte and macrophage dynamics during atherogenesis. Arterioscler Thromb Vasc Biol. 2011 Jul;31(7):1506-16.
Material and Methods
The authors have examined human tissues obtained during carotid endarterectomy or aortic valve surgical replacement and all patients provided informed consent. However, we know nothing about these patients (even age and sex is not given). These and other basic clinical data are important to make a valuable comparison of the presented findings with other publications' results. The authors seem to be aware of this fact (lines 58-63). Were there any exclusion criteria, for example, bicuspid valve morphology, rheumatic heart disease, endocarditis? In this reviewer opinion, examined groups must be more closely described. There is no information, how were selected valve parts for histological analysis. This is important, since it is known that in the stenotic valve leaflets are areas differently affected and almost intact.
Results
Results should provide, first of all, the authors findings. In the current form, approximately half of the results section reports literature data addressed by the authors own results/images. There are also some speculative statements, not supported by the results. They might be placed in Discussion but not in Results (e.g. lines 87-89; 106-108; 167-172). The design of this study makes it impossible to prove the sequences of events suggested by the authors.
Lines 104-106
The division into canonical (without inclusions) and non-canonical (with granules) macrophages are not clear. These terms are used in the case of macrophages, usually in context to their activation mode.
Lines 100-102
“The majority of the macrophages had elliptic or round large nuclei and did not have any specific inclusions in the cytosol, suggesting their moderate activity in the ECM remodeling..” It is not clear if in the atherosclerotic plaques this form was also predominant? In lines 83-84 is stated “…foam cells were the most abundant in the atherosclerotic plaques…”.
Discussion
Line 205
Presented references are related to Ischemic Cardiovascular Disease, not generally, to CVD.
Lines 218-219
“However, the macrophage diversity in plaques (and also in dysfunctional native valves) was limited, whereas BHVs contained a number of macrophage appearances.” It is not clear what kind of appearances? In Results is given only “These non-canonical macrophages with cytoplasmic granules were prominent in BHVs” (lines 105-106).
Blood vessels are discussed twice in this section (lines 219-224 and 265-267), however, based practically only on the authors findings. This is surprising since there are reports not corroborating the author’s claim that blood vessels in degenerating aortic valves are leaky. [Akahori H et al. Eur Heart J. 2011 Apr;32(7):888-96.]
How to explain that macrophages with electron-dense granules, containing, as the authors speculate, internalized haemoglobin, were prominent in BHV and not present (?) in atherosclerotic plaques, in which haemorhages were seen as well. On the other hand, one would expect the presence of internalized erythrocyte “ghosts” in some macrophages. Did you see any?
Figures
Figures 1, 2 4,5, 6
“Native heart valves” should be replaced with a “Native aortic valve”.
Figure 1B
In the reviewer's opinion, aortic valve micrograph presents the rim of large calcification and lipid accumulation in neighboring tissue, which should not be called “fatty steaks”.
Figure 2
There is a lack of three micrographs (B-atherosclerotic plaque; C and D – native aortic valve). This is probably due to the wrong pdf conversion. However, since the caption explains that all those missing pictures are associated with the absence of examined cells in the respective tissue, maybe this is the author's decision to leave the empty parts in this figure.
Author Response

(The authors gave the same response as above.)

Reviewer 3 Report
The manuscript of Kostyunin et al. investigated ultrastructural features of calcified cardiovascular tissues including atherosclerotic plaques, calcified aortic valves and bioprosthetic heart valves. They addressed whether lipid retention, macrophage infiltration, intraplaque/intraleaflet hemorrhage, and calcification are common or unique characteristics in these different tissues.
This is a purely observational study using electron microscopy to show the common and different features of cardiovascular tissues. The results are nicely presented and discussed. The authors summarize the findings in a nice schematic.
My only concern regarding this work is that I am not sure whether this paper fits into this specific journal.
Author Response
We sincerely thank the reviewer for the evaluation of our study.
Round 2
Reviewer 2 Report
Submitted by the authors, the revised version of the manuscript is much improved. However, some problems still remain unsolved.
Lines 79-80
The authors claim that backscattered SEM has never been applied in cardiovascular pathology, while in the same sentence, they present contradicting references (20,21).
Line 222 – 234, 272,314, Figure 7.
In my previous revision, I have stated that some references are related to Ischemic Cardiovascular Disease, not generally, to CVD (line 205 at an earlier manuscript). The authors have changed the references. However, my concern is to the term CVD used by the authors. According to the World Health Organization definition, Cardiovascular Diseases (CVDs) are a group of disorders of the heart and blood vessels, including coronary heart disease, cerebrovascular disease, peripheral arterial disease, rheumatic heart disease, congenital heart disease, deep vein thrombosis and pulmonary embolism. Some of them were out of the study's scope while others intentionally excluded (bicuspid AV, rheumatic heart disease). In my opinion, the authors should not address their results to CVD in general, but only to those examined. First of all, in the central figure (Fig. 7) description “Ultrastructural features of CVD” must be changed. If not, the authors should explain how and why they define CVD or why they claim that this figure can be related to all pathologies commonly described as cardiovascular diseases.
Figure 2
The concept of putting gray/empty images representing the lack of particular ultrastructural macrophage phenotypes seems strange to me and has never been seen before. In most papers, the absence of a specific feature in examined material is shown in the micrograph as well. However, since these are EM images (low areas due to relatively high magnifications), maybe a better option to avoid the blank pictures is to rearrange or divide the whole figure?
Author Response
We sincerely thank the reviewer for the constructive criticism and valuable comments.
Reviewer 2
Submitted by the authors, the revised version of the manuscript is much improved. However, some problems still remain unsolved.
Lines 79-80
The authors claim that backscattered SEM has never been applied in cardiovascular pathology, while in the same sentence, they present contradicting references (20,21).
We replaced the references, yet Sergio Bertazzo and colleagues exploited another technique, superimposing SEM and BSEM images and therefore obtaining different images. Also, they investigated calcium phosphate transformation (i.e., chemical properties) rather than histopathological features. Nevertheless, we generally agree with the reviewer that the previous references could confuse the reader.
Line 222 – 234, 272,314, Figure 7.
In my previous revision, I have stated that some references are related to Ischemic Cardiovascular Disease, not generally, to CVD (line 205 at an earlier manuscript). The authors have changed the references. However, my concern is to the term CVD used by the authors. According to the World Health Organization definition, Cardiovascular Diseases (CVDs) are a group of disorders of the heart and blood vessels, including coronary heart disease, cerebrovascular disease, peripheral arterial disease, rheumatic heart disease, congenital heart disease, deep vein thrombosis and pulmonary embolism. Some of them were out of the study's scope while others intentionally excluded (bicuspid AV, rheumatic heart disease). In my opinion, the authors should not address their results to CVD in general, but only to those examined. First of all, in the central figure (Fig. 7) description “Ultrastructural features of CVD” must be changed. If not, the authors should explain how and why they define CVD or why they claim that this figure can be related to all pathologies commonly described as cardiovascular diseases.
We revised the Figure 7 according to the reviewer’s comments and removed the term “CVD” from the manuscript, mentioning strictly the disorders we examined in this paper (atherosclerosis, CAVD and SVD).
Figure 2
The concept of putting gray/empty images representing the lack of particular ultrastructural macrophage phenotypes seems strange to me and has never been seen before. In most papers, the absence of a specific feature in examined material is shown in the micrograph as well. However, since these are EM images (low areas due to relatively high magnifications), maybe a better option to avoid the blank pictures is to rearrange or divide the whole figure?
We replaced the gray (blank) images with the respective EM images but without arrow annotations, denoting the absence of the respective macrophage populations.
We sincerely thank the reviewer for the constructive criticism and valuable comments.